# Comparative Transcriptome Analysis of Grafted Tomato with Drought Tolerance

**DOI:** 10.3390/plants11151947

**Published:** 2022-07-27

**Authors:** Maria Isabel Fuentes-Merlos, Masaru Bamba, Shusei Sato, Atsushi Higashitani

**Affiliations:** Graduate School of Life Sciences, Tohoku University, Sendai 980-8577, Japan; masaru.bamba.b2@tohoku.ac.jp (M.B.); shusei.sato.c1@tohoku.ac.jp (S.S.)

**Keywords:** tomato, grafting, drought stress, transcriptomics, phytohormones

## Abstract

Grafting is a method used in agriculture to improve crop production and tolerance to biotic and abiotic stress. This technique is widely used in tomato, *Solanum lycopersicum* L.; however, the effects of grafting on changes in gene expression associated with stress tolerance in shoot apical meristem cells are still under-discovered. To clarify the effect of grafting, we performed a transcriptomic analysis between non-grafted and grafted tomatoes using the tomato variety Momotaro-scion and rootstock varieties, TD1, GS, and GF. Drought tolerance was significantly improved not only by a combination of compatible resistant rootstock TD1 but also by self-grafted compared to non-grafted lines. Next, we found the differences in gene expression between grafted and non-grafted plants before and during drought stress treatment. These altered genes are involved in the regulation of plant hormones, stress response, and cell proliferation. Furthermore, when comparing compatible (Momo/TD1 and Momo/Momo) and incompatible (Momo/GF) grafted lines, the incompatible line reduced gene expression associated with phytohormones but increased in wounding and starvation stress-response genes. These results conclude that grafting generates drought stress tolerance through several gene expression changes in the apical meristem.

## 1. Introduction

Drought stress causes damage to crop plants, generating economic and production losses. Grafting is a method used in agriculture that improves crop production and biotic and abiotic stress tolerance through water and nutrient uptake enhancement [1,2]. This technique consists in connecting the shoot or “scion” and the root or “rootstock”, ensuring that the compatibility of the graft combinations is successful [3].

Previous studies have established critical parameters for graft formation, including cell adhesion, division, proliferation, and vascular reconnections [4]. In addition, plant hormones such as auxins, cytokinin, and gibberellins, play an essential role in cell proliferation at the graft junction, aiding in the correct progression of vascular reconnection mechanisms. Auxin plays an essential role in cell proliferation and vascular reconnection in the graft junction [5,6]. Ethylene and cytokinin have a minor contribution to the phloem reconnection [5]. Gibberellin is involved in expanding cortex cells in the vascular tissue [6]. Several studies have elucidated graft union formation; however, research on stress-tolerance mechanisms is still in progress.

Grafted *Solanaceae* and *Cucurbitaceae* crop plants have been recently used for grafting, becoming essential for stress-tolerance studies [7]. For example, in tomato (*Solanum lycopersicum*) plants, many studies have reported improved fruit yield, quality, and resistance to biotic and abiotic stress [1,7,8,9]. Severe drought stress causes a reduction of photosynthesis rate and root activity, and, under mild stress, water deficiency increases total carotenoid and vitamin C content in the tomato fruit [10,11]. The reduction of photosynthesis is related to the abscisic acid (ABA) signaling pathway, where this hormone regulates the stomatal closure, decreasing the CO_2_ uptake in leaves [12]. ABA activity activates a signaling pathway that triggers reactive oxygen species (ROS) production. As a result, several antioxidant systems are activated, such as ROS scavenger enzymes and antioxidant molecules, such as ascorbate peroxidase (APX) and vitamin C [13,14].

Momotaro tomato varieties, including “Momotaro peace” (Momo), are bred and cultivated in greenhouse and fields in Japan and are mainly known for their sweetness, hardiness, and flavor [15]. In addition, grafts of this variety have been widely used to increase biotic and abiotic stress tolerance and tomato fruit quality and yield [16]. However, the molecular basis and effects of grafting on drought stress tolerance are still under investigation. Therefore, this study presents transcriptional alterations on shoot apical meristem by grafting. In addition, we assessed whether vigorous rootstock and grafted Momo could improve drought tolerance by transcriptomic analysis between grafted and non-grafted lines.

## 2. Results

### 2.1. Identification of Tissue Types Involved in the Recovery of Drought-Tolerant Grafted Lines

To determine the effect of grafting on drought stress in the Momo shoot apical meristem, two different homo-grafted lines, Momo-scion and green force rootstock (Momo/GF) and Momo-scion and TTM-079 rootstock (Momo/TD1), and a self-grafted line (Momo/Momo) were used as compared with the non-grafted lines, GF, TD1, green guard (GS), and Momo. All lines were subjected to 12 d of drought stress treatment by withholding water (Figure 1a). Only the Momo/TD1 plants demonstrated a 100% recovery rate, whereas the control lines had a recovery ratio of less than 40% (Figure 1b). Interestingly, the self-grafted Momo/Momo were more resistant to drought stress (56% recovery ratio) than non-grafted Momo (20% recovery ratio). It was shown that grafting results in drought stress tolerance regardless of root varieties. Additionally, these grafted lines showed recovery from apical meristems or axillary buds, which were thought to have reduced their apical dominance (Figure 1c). On the other hand, the homo-grafted Momo/GF decreased resistance (less than 20% from 40% of GF control) and showed no recovery from the axillary buds, which could have been due to graft incompatibility.

### 2.2. Differences in Gene Expression between Grafted and Non-Grafted Plants before and during Drought Stress Treatment

To identify genes whose expression was altered by grafting and resulted in changes in drought tolerance, we performed RNA sequencing (RNA-seq) on shoot apical meristem before (D0) and 3-day drought stress treatment (D3). First, counts per million (CPM) values for gene expression were calculated between the formed clusters of the control and grafted lines and between D0 and D3 (Figure 2a). Subsequently, a heatmap was generated using the control and grafted gene expression data, in which we observed two major clusters, drought and grafting effect. The clusters separated the control and grafted lines, and D0 and D3 treated lines are shown in the upper part of the heatmap (Figure 2b).

Comparison between grafted and non-grafted control group under D0 revealed that grafting caused down-regulation (Log_2_ fold change ≤ −1) of 602 differentially expressed genes (DEGs) and up-regulation (Log_2_ fold change ≥ 1) of 1158 DEGs (Figure 3a, Appendix A). The classification of these DEGs by gene ontology (GO) analysis indicated that several up-regulated genes were related to the following categories: “carbohydrate metabolic process: 23 DEGs”, “gene expression: 4 DEGs”, “response to stimulus: 56 DEGs”, such as “temperature stimuli (heat shock protein): 14 DEGs”, “response to reactive oxygen species (ROS): 8 DEGs”, and “response to abscisic acid: 9 DEGs” (Figure 3b). GO analysis of the down-regulated genes was associated with different categories, namely “nucleic acid metabolic process: 42 DEGs”, “RNA modification: 25 DEGs”, “chloroplast organization: 7 DEGs”, “positive regulation of helicase activity: 4 DEGs”, and “mitochondria DNA replication: 4 DEGs”.

We also listed several hormone-related genes that were differentially expressed between grafted and control plants in Table 1. Those hormone-related genes include ABA receptors, auxin, cytokinin, ethylene-responsive transcription factors, and gibberellins (GA). Several small auxins up-regulated RNAs (SAURs), dormancy-associated gene/auxin-repressed protein, auxin efflux carriers, and auxin response factors were found in the auxin classification. Additionally, cytokinin was differentially expressed in cytokinin biosynthesis pathway enzymes. Grafting differently regulated ethylene-responsive transcription factors and response factors, with the majority being up-regulated. A similar pattern was observed with the gibberellin-regulated protein (GASA).

DEGs obtained by comparing the gene expressions in D3 and D0 were used to evaluate the effects of drought stress in the control and grafted lines. Transcriptomic analysis revealed that 3936 DEGs were down-regulated in D3 while 2903 were up-regulated in the control lines (Figure 4a). Similar patterns, but slightly decreased, were observed in the grafted lines, with 3304 and 2542 DEGs down- and up-regulated, respectively (Figure 4b). Additionally, DEGs classified by GO analysis showed gene regulation similarities in grafted and control lines. However, control lines exhibited more genes in each GOs category (Figure 4c) than those in the grafted lines. The commonly up-regulated (control/grafted) are related to the “biological regulation: 254/191 DEGs”, “response to stimulus: 187/104 DEGs”, “catabolic process: 143/97 DEGs”, “regulation of RNA metabolic process: 121/107 DEGs”, “proteolysis: 80/58 DEGs”, “protein ubiquitination: 47/29 DEGs”, and “positive regulation of transcription: 34 DEGs”. The commonly down-regulated were identified as “cell division: 16/12 DEGs”, “photosynthesis, light-harvesting: 18/9 DEGs”, “fatty acid biosynthesis process: 19/9 DEGs”, “cell wall organization or biogenesis: 32/25 DEGs”, “RNA modification: 45/4 DEGs”, “DNA replication: 46/34 DEGs”, “cell cycle: 126/100 DEGs”, and “response to stimulus: 224/181 DEGs”. These results indicate that drought stress has a more significant impact on the control lines than the grafted lines.

We performed a comparative analysis between the control and grafted lines under D3 to clarify the observed differences between DEGs due to grafting. Gene expression analysis revealed that grafting resulted in up-regulated 1117 DEGs and down-regulated 518 DEGs in comparison to control (Figure 4d, Appendix A). Several of the up-regulated genes are related to the “cell cycle: 42 DEGs”, “DNA repair: 17 DEGs”, “DNA replication: 14 DEGs”, “reproduction: 12 DEGs”, “photosynthesis: 10 DEGs”, and “shoot system development: 7 DEGs” categories. Down-regulated DEGs were identified as a “response to abiotic stimulus: 13 DEGs” that included “water deprivation: 5 DEGs” and “temperature stimulus: 9 DEGs” (Figure 4e). These results show that non-grafted and grafted lines have several DEG, further supporting observational analysis results in which grafted lines continued to have normal growth patterns and were less affected by the stress stimulus. By contrast, the control lines were heavily affected and could not withstand drought stress conditions beyond day 3.

To confirm RNA-seq results, we analyzed expression levels of 4 genes, heat shock protein 20 (*HSP20*), ascorbate peroxidase (*APX*), chlorophyll a–b binding protein (*LHCB*) and 9-cis-epoxycaratenoid dioxygenase (*NCED*) by real-time quantitative PCR (Figure 5). In the grafted lines, *HSP20* and *APX* showed up-regulation at D0, but their expression was reduced or unchanged at D3. In the control lines, their lower expression at D0 significantly increased at D3 (Figure 5a,b). By contrast, *LHCB* showed down-regulation under D3 in the control and grafted lines; however, in the grafted lines, the decrease in gene expression was less severe than in the control lines (Figure 5c). *NCED* gene expression showed significant up-regulation in response to drought stress (D3) in several lines (GF, Momo, Momo/TD1 and Momo/Momo) (Figure 5d). Interestingly, these up-regulations were commonly observed in lines with high survival rates to drought stress treatment (Figure 1b).

Furthermore, when comparing compatible (Momo/TD1 and Momo/Momo) and incompatible (Momo/GF) grafted lines, the DEGs (Appendix A) in which GO is commonly enriched in both compatible lines were related to “regulation of transcription: 33 DEGs”, “signaling: 13 DEGs”, “response to hormone: 12 DEGs”, and “response to abiotic stimulus: 8 DEGs” categories (Figure 6a, Table 2). By contrast, high expression in incompatible line showed GO enrichment genes in different categories such as “response to wounding: 2 DEGs”, “plant-type cell wall biogenesis: 3 DEGs”, “phospholipid metabolic process: 3 DEGs”, and “response to stress: 7 DEGs” (Figure 6b, Table 2). These possibly lead to activation of pathway differentiation related to graft junction compatibility, suggesting that grafting enhances the drought-tolerance effect in the combination of scion and rootstock.

## 3. Discussion

In general, grafting can improve plant development and stress tolerance [1,2]. Our results show that Momo/TD1 has a high survival rate (Figure 1b) indicating that this combination is the most successful for drought tolerance in Momo-scion. TD1 has been selected and used as one of vigorous rootstocks by Takii Seed Co., Ltd. However, ungrafted TD1 was less tolerant, indicating that grafting is needed to acquire stress tolerance. In addition, unexpectedly, self-grafted Momo/Momo was found to be more resistant to drought than non-grafted Momo (Figure 1b). This suggests that the process of cutting and connecting in the graft activates several wound-healing mechanisms, leading to increased drought tolerance. Wound stress can activate several responsive signaling at the wound sites and can be propagated to the rest of the plant [17,18]. Similarly, this signaling can pass between the rootstock and scion, such as reactive oxygen species (ROS), phytohormones, metabolites, and genetic information as small RNAs [2,19]. Recently, a study in grafted junction of tomato seedlings showed differentially expressed genes (DEGs) related to several pathways, such as oxidative stress and hormones, ABA, ethylene, auxin, gibberellin, and jasmonic acid [20]. Such ROS signaling and hormonal changes at the grafted site can affect the apical meristem of the scion.

As ROS signaling, HSPs and ROS scavenger enzymes increased in the apical meristem of the grafted lines before drought stress (D0) (Appendix A, Figure 3b and Figure 5a,b), these pre-activations can lead to drought stress tolerance. In fact, some studies on the overexpression of HSPs and APX have shown increased tolerance to drought stress [21,22,23,24]. Furthermore, these activations are regulated by ABA signaling pathway in plants [25,26]. For example, HSP70 in cucumber (*Cucumis sativus* L.) was induced by a stress-tolerant rootstock pumpkin (*Cucurbita moschata*) and luffa (*Luffa cylindrica Roem.*) through ABA-dependent activation [27,28]. As the result shown in Table 1 and Figure 3b, the graft strongly activated several ABA-related genes. ABA is a vital rapidly produced stress-responsive hormone that can help increase the plant’s survival during drought stress [10,29]. ABA signaling involves ABA receptor PYL [Pyrabactin Resistance 1 (PYR)/pyrabactin resistance-like (PYL)], which generates a cascade of response and signal transduction [10]. The activation of ABA receptors PYL in the shoot apical tissue could be due to water stress caused by the disconnection of the vascular system.

The SAURs family is one example of an auxin-responsive gene family. SAURs are auxin-induced and play a role in cell elongation. Additionally, they can act independently of auxin, regulated by other hormones, transcription factors, and environmental stress [30]. Some SAURs (*Solyc07g042490*, *Sl-SAUR5*, and *Sl-SAUR26*) were up-regulated under D0 (Table 1), whereas their Arabidopsis (*Arabidopsis thaliana*) orthologs (*SAUR76*, *SAUR8*, and *SAUR 14*) are regulated by auxin, ethylene, or environmental conditions, including light [31,32,33,34]. By contrast, the down-regulation of *Sl-SAUR55* (*Solyc05g046320*) is related to leaf senescence, dormancy/auxin associated protein (*Solyc01g099840*), and auxin-repressed protein (*Solyc02g077880*) associated with pathogen response, which are triggered by other conditions other than auxin [35,36,37]. Based on these results, we can assume that auxin activated several genes in the shoot apical meristem of Momo-scion, which can regulate stress tolerance in association with other hormones [38].

Cytokinin appears to be present in the shoot apical meristem of the Momo-scion, where biosynthetic genes (*Solyc06g075090*, *Solyc04g080820*, and *Solyc10g082020*) are up-regulated, and *SlCRF4* (*Solyc03g007460*), which is not induced by cytokinin, is down-regulated [39]. Additionally, ethylene, essential for abiotic stress response, was up-regulated by grafting (Table 1). We identified several ethylene-responsive transcription factors that were up-regulated by grafting. Therefore, we expected interaction between ethylene and auxin for stress tolerance, as it promotes vascular cell division during graft union formation [40,41]. GA-regulated proteins are usually activated by GA. Several genes have been found to play a role in cell growth and differentiation and the ABA signaling pathway [42]. Several days after grafting (Table 1), GA was not activated in grafted plants, where GA-regulated proteins 3, 14 and *Solyc12g042500* (ortholog of GASA11, 14, and 10 in Arabidopsis) are not affected by GA or can be inhibited by exogenous GA application [42,43]. However, GASA14 (an ortholog of *Solyc03g113910*) was induced by ABA treatment [43].

During D3 of drought stress treatment, the overall change in gene expression by grafting (both up-regulation and down-regulation) was slightly reduced compared to the control (Figure 4a,b). On the other hand, grafting at D3 increased about 1000 genes and decreased about 500 genes compared to control (Figure 4d). Interestingly, some of these genes that are apparently increased (GOs: cell cycle, DNA replication, and photosynthesis) are down-regulated by drought, and conversely, some of these genes that are apparently decreased (GOs: response to abiotic stimulus) are up-regulated (Figure 4c,e). This suggests that grafting reduces the drought response by suppressing the activation or reduction of gene expression.

Furthermore, Momo-scion’s sensitivity to drought depended on the rootstock used (Figure 1b). The interaction between scions and rootstocks is essential for the phenotypic variability of grafted plants [2]. The drought sensitivity of the grafted Momo/GF could be related to the incompatibility between the scion and rootstock. When the separated tissues are attached, and incomplete reconnection of the xylem occur during vascular reconnection, the root and shoot water potential can be affected [5,44,45]. We observed differences in gene expression between the compatible (Momo/Momo and Momo/TD1) and incompatible lines (Momo/GF), such as the “response to hormones” and “response to stress” (Figure 6a,b, Table 2). Hormonal dynamics contribute graft reconnection and communication between the rootstock and the scion [6,46]. One possible reason for drought tolerance in the compatible and incompatible lines could be related to a rapid physiological response to the stress, such as response to ABA.

An ABA receptor *PYL4*-like was induced in compatible lines as compared to Momo/GF (Table 2). Likewise, it was observed that an essential enzyme in the ABA biosynthetic pathway, 9-cis-epoxycarotenoid dioxygenase (*NCED*), was up-regulated on D3 in the compatible lines (Figure 5d). *NCED* plays a role in the stomatal closure, generating a decrease in gas exchange and photosynthesis [14]. Moreover, our results showed that during drought stress, grafting had less reduction in the expression of photosynthesis-related genes than in the control (Figure 4c,e). For example, chlorophyll a–b binding proteins (*LHCB*) are part of the early light-induced protein, with the primary function as photosynthetic light-harvesting complex but also as stress-responsive genes, especially under drought stress [47,48]. An *LHCB* (*Solyc08g067330*) in our study was reduced during drought stress in control and lesser reduced in the grafted lines (Figure 5c). The stress-responsive genes activated in Momo/GF could demonstrate that oxidative stress occurring in the scion, which could indicate a lack of vascular reconnection [49]. Nevertheless, the main reasons for the incompatibility between same or different graft species are largely unknown.

Communication signals between scion and rootstock are important for stress response and survival of tomato. One hypothesis of how grafting alters gene expression in the shoot apical meristem after several days of the graft-healing process could be through epigenetic modifications. We are attempting to study those epigenetic changes in Momo-scion.

## 4. Materials and Methods

### 4.1. Plant Materials, Growth Conditions, and Grafting

First-generation (F1) hybrid green force (GF), green guard (GS), TTM-079 (TD1), and Momotaro (Momo) tomato seeds (Takii Seed Co., Ltd., Kyoto, Japan) were sown in soil with vermiculite in a 30-well strip (5 × 5 cm per well) and placed in a large growth cabinet (Espec Ltd., Osaka, Japan), at 28/22 °C (day/night), with a 12 h photoperiod (300 μmol/m2/s) and 85/75% relative humidity. Thirty germinated seedlings of each line were watered every day to keep the soil moisturized without waterlogging.

For grafting, three different graft combinations were made using the same Momo as the scion [homo-grafted (Momo/TD1 and Momo/GF), and self-grafted (Momo/Momo)]. Before grafting (3 true leaves stage), the stem diameter below the hypocotyl should be 1.7–2.2 mm for the rootstock and scion before grafting. Then, a diagonal cut was made using a graft-cutter, and the scion with the rootstock was joined using a joint graft holder (Seem, Kita-Kyushu, Japan). The grafted material was kept under high humidity (85–90%) and low light for 3 d during the healing process. After that, the humidity was gradually decrease and light irradiation increased. After 5 d, the grafted plants were acclimated to growth cabinet conditions. A total of 3 weeks after grafting, plants were used for further experiments.

### 4.2. Drought Stress Treatment and Sample Collection

Ungrafted lines referred to as control (2-week-old plants with 4 true leaves), and grafted plants (3 weeks after grafting with 5 true leaves), were subjected to drought stress by withholding water for 12 d and then irrigated for 3 d to measure the survival rate (Figure 1a). The plants were kept in a 30-well strip per line in the same growth cabinet under the same conditions as described previously. D0 is before applying the stress and D3 is withholding water for 3 days. The shoot apical meristems (bud area containing the apical meristem and young leaves) of each line were collected in three biological replicates at different time points (D0 and D3).

### 4.3. RNA Extraction and RNA Sequencing and Data Analysis

Total RNA was extracted from shoot apical meristematic tissues of lines exposed to D0 and D3 using TRIzol reagent (Thermo Fisher Scientific, Waltham, MA, USA) and RNeasy Mini Kit (Qiagen, Hilden, Germany) for genomic DNA elimination. RNA quantity was obtained by measuring by absorbance using a Quantifier RNA NanoQuant Infinite 200 (TECAN, Mannedorf, Switzerland). RNA integrity was verified using an Agilent 2200 TapeStation System (Agilent Technologies, Santa Clara, CA, USA).

RNA pools of the three biological replicates per line were used for all RNA sequencing experiments. Samples were purified using a Ribo-Zero rRNA Removal Kit [Plant Leaf (Illumina, San Diego, CA, USA)]. Furthermore, the SureSelect Strand-Specific RNA Library Prep for Illumina Multiplexed Sequencing (Agilent Technologies) and NextSeq 500 (Illumina) were used for mRNA library preparation. The sequencing service was provided by the Kazusa DNA Research Institute, Chiba, Japan. All data analysis was performed as described in Refs. [50,51,52,53] with ITAG4.0 tomato annotation information [54] and featureCounts v.2.01 [55].

Raw read counts were normalized to Reads per Kilobase per Million (RPKM) to determine gene expression. For DEGs, raw read counts were used and analyzed using the limma-voom package [56] to generate normalized values as logarithm count per million (log-CPM). We chose to examine counts per million (CPM) instead of reads per kilobase per million (RPKM) [57] because of our interest in comparing relative changes in expression between conditions. Filtered genes expressed above 0.5 CPM in at least 1 sample in control and grafted lines were retained. Log-CPM values were transformed to log2 fold change between the non-grafted as a control and grafted as a treatment, to measure the grafting effect. Additionally, D0 was used as a control and D3 was used as a treatment to measure the drought effect.

### 4.4. Real-Time Quantitative PCR (RT-PCR) in Stress-Responsive Genes

RT-PCR was performed with PrimeScript RT Reagent Kit (Takara Bio Inc., Shiga, Japan) for reverse transcription to cDNA. For each reaction, 4 μL of cDNA (1:100 dilution) and 5.8 μL of KAPA SYBR FAST universal (Kapa Biosystems Inc., Wilmington, MA, USA), and 0.2 μL of primer mix were used. CFX Connect Real-Time System (Bio-Rad Laboratories, Hercules, CA, USA) with the following cycling condition: 3 min at 95 °C, followed by 40 cycles of 95 °C for 15 s, 60 °C for 30 s, and 72 °C for 1 min. Three biological replicates with three technical repetitions were tested. The housekeeping gene 18S rRNA was set as the endogenous control, where forward 5′-ATGATAACTCGACGGATCGC-3′ and reverse 5′-CTTGGATGTGGTAGCCGTTT-3′. The selected genes used to verify the results of the RNA-seq data were heat shock protein 20 (*HSP20*), ascorbate peroxidase (*APX*), chlorophyll a–b binding protein (*LHCB*), and 9-cis-epoxycaratenoid dioxygenase (*NCED*). For *HSP20* (*Solyc08g062450*), the forward 5′-CCGGTGAAGATTCCGACAAG-3′ and reverse 5′-TTCACATCCGCTGGTGTAGC-3′ was used. For *APX* (*Solyc09g007270*), the forward 5′-TCAGGCACCCGAATGAACTT-3′ and reverse 5′-GGGCCTCCCGTAACTTCAAC-3′ was used. For *LHCB* (*Solyc08g067330*), the forward 5′-GGGCCTGACCGTGTGAAGTA-3′ and reverse 5′-AGTCCAGCAGTGTCCCATCC-3′. Lastly, for *NCED* (*Solyc07g056570*), the forward 5′-GCTGGAATGGTGAACCGAAA-3′ and reverse 5′-TGCTGTTGGGGTCTCTTGGT-3′. The relative expression of each gene was calculated with the 2^−^^△C’T^ method [58].

### 4.5. Gene Ontology Categorization

The PANTHER 14.0 tool [59] was used for the GO enrichment analysis of the DEGs with false discovery rate (FDR) < 0.05 and *p*-value < 0.05. Only GOs categories of interest were selected from the biological process family.

## 5. Conclusions

Our results show that grafting, especially self-grafting, increases drought stress survival rate by inducing stress-adaptative mechanisms on transcriptomic changes on the shoot apical meristem of Momo. Additionally, the drought resistance is highly dependent on the grafting compatibility and rootstock used.

## Figures and Tables

**Figure 1 plants-11-01947-f001:**
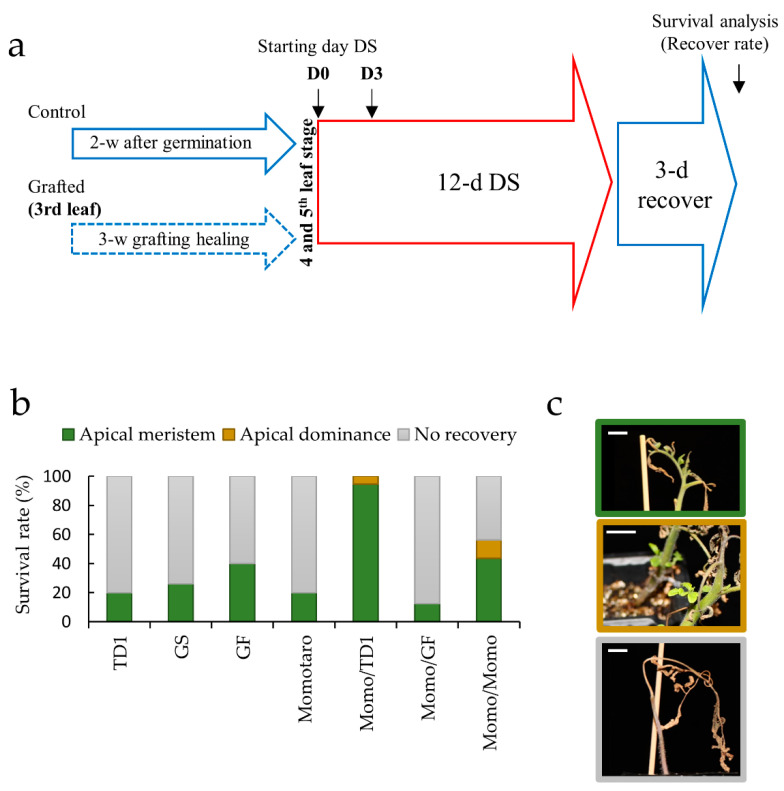
Grafting in tomato plants increases survival rate through the shoot apical meristem and apical dominance recovery from the axillary bud. (**a**) Schematic representation of the drought stress experimental design, marking the points of sample collection for RNA sequencing. D0 indicated before treatment and D3 indicates 3 d of drought stress treatment. GH = grafting healing, DS = drought stress. (**b**) Survival rates of all tomato lines after drought stress treatment. (**c**) Phenotype of self-grafted Momotaro (Momo/Momo) tomato recovery after 12 d of drought stress treatment through apical meristem, apical dominance, or no recovery. Control: TTM-079 [TD1 (*n* = 30)], green guard [GS (*n* = 30)], green force [GF (*n* = 30)], and Momo (*n* = 30). Grafted: Momo/TD1 (*n* = 19), Momo/GF (*n* = 8), and Momo/Momo (*n* = 16). Green = apical meristem, brown = apical dominance, and gray = no recovery. White scale = 1 cm.

**Figure 2 plants-11-01947-f002:**
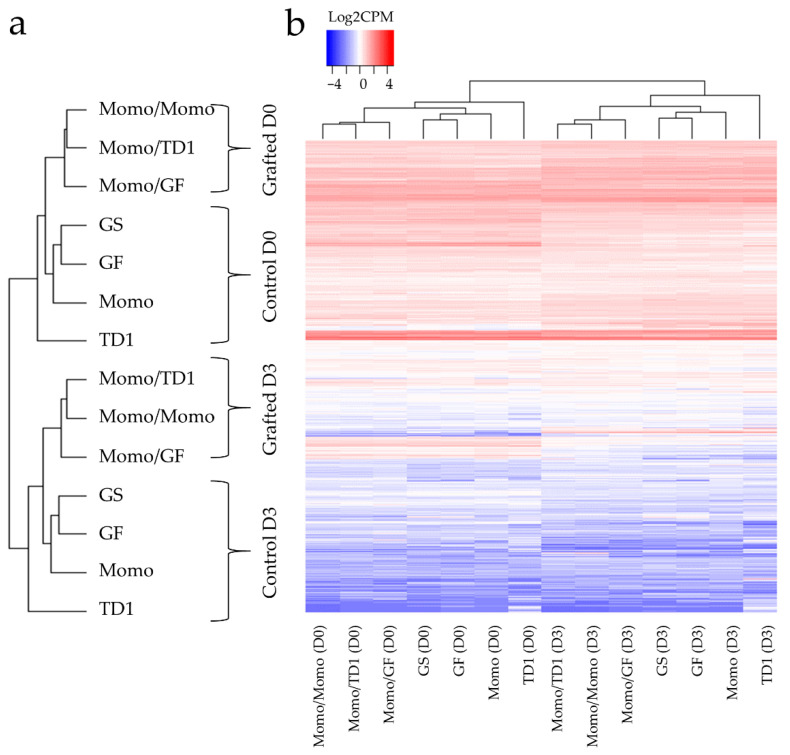
Control and grafted tomato lines show different gene expression pattern under normal and stress conditions. (**a**) Dendrogram of control and grafted tomato lines under D0 and D3. Brackets indicated sample lines that were grouped as control and grafted, and D0 and D3. (**b**) Heatmap of the gene expression in logarithm counts per million (Log_2_ CPM) of the different grafted combinations and non-grafted tomato lines. Red and blue indicate high and low expressions, respectively. Top brackets group the gene expression similarities between tomato lines. D0 and D3 indicates before drought stress and day 3 during drought stress treatment.

**Figure 3 plants-11-01947-f003:**
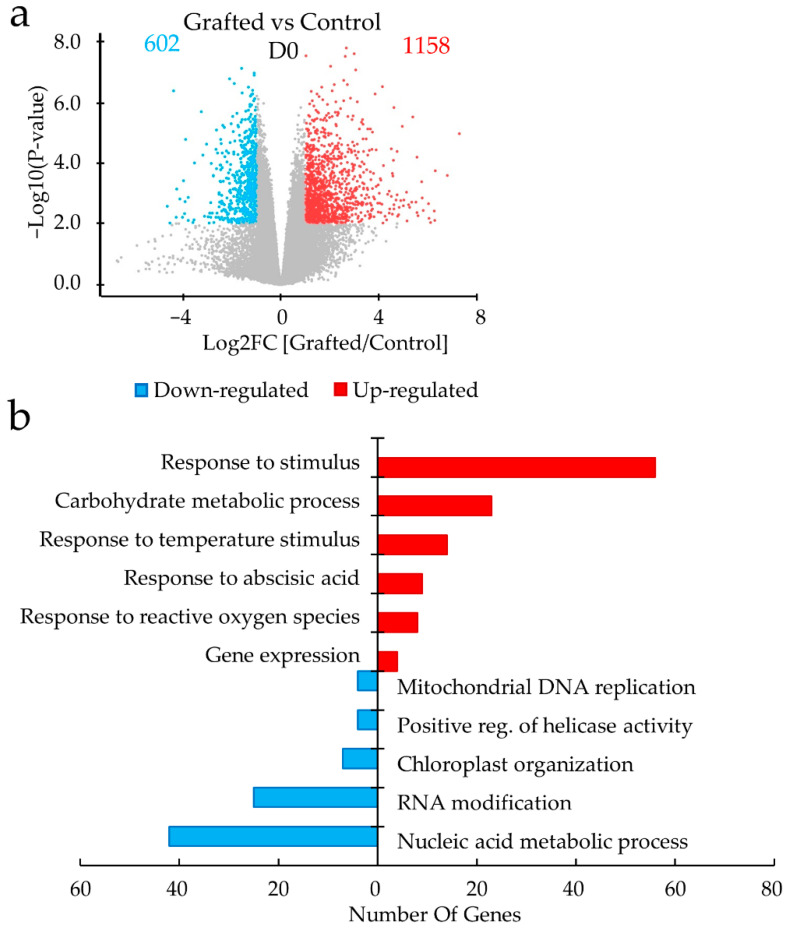
Comparison of the differentially expressed genes (DEGs) in grafted and control before drought stress treatment. (**a**) Volcano plot representation of DEGs up-regulated and down-regulated by grafting. The *x* axis shows Log_2_ fold change (FC) between Grafting vs. Control and *y* axis shows −Log10 (*p*-value). (**b**) Bar plot showing gene ontology (GO) enrichment analysis (Panther, false discovery rate < 0.05) for biological process in DEGs by grafted plants. Red and blue color indicate high and low gene expressions on significant DEGs (Adj. *p* < 0.05, Log_2_FC ≥ |1|). The *x* axis shows the number of genes categorized in each GO term (*y* axis).

**Figure 4 plants-11-01947-f004:**
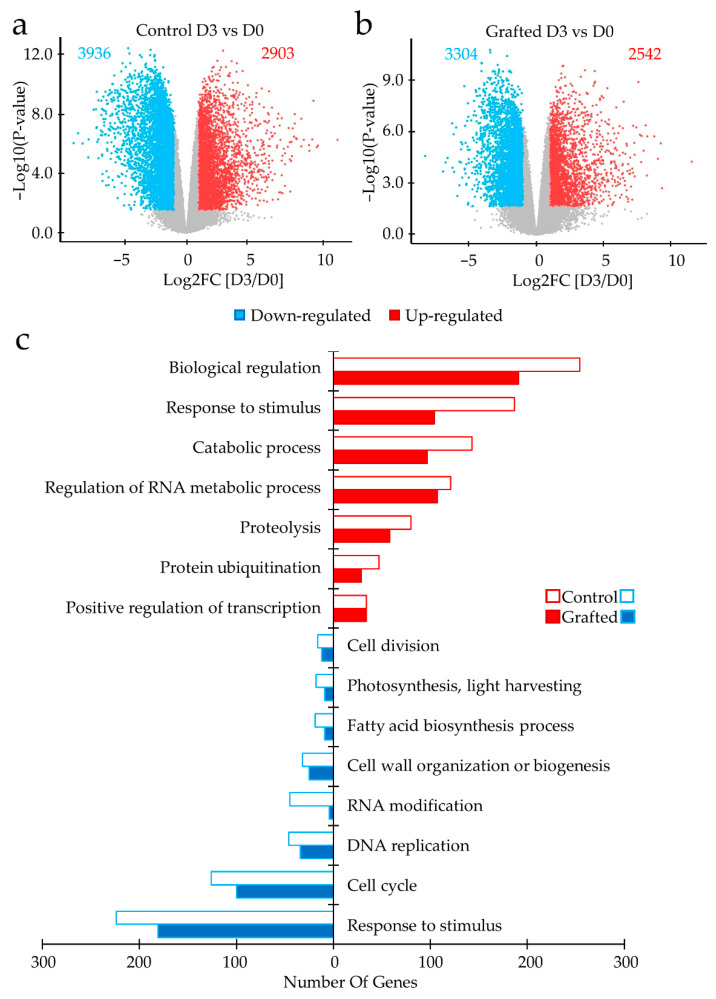
Differentially expressed genes (DEGs) and enriched pathways in grafted and control shoot apical meristem before drought stress and day 3 during stress. Volcano plot representation of DEGs up-regulated and down-regulated by drought stress treatment in (**a**) control and (**b**) grafted samples. The *x* axis shows logarithm fold change (Log2FC [D3/D0]) and *y* axis shows –Log10 (*p*-value). Significant DEGs (Adj. *p* < 0.05, Log2FC ≥ |1|) are in red and blue for up-regulation and down-regulation. (**c**) Bar plot showing gene ontology (GO) enrichment analysis (Panther, false discovery rate (FDR) < 0.05) for biological processes on significant DEGs of the control and grafted samples. Empty bars indicate the control and fill bars indicate the grafted samples. The *x* axis shows the number of genes categorized in each GO term (*y* axis). (**d**) Comparison of the DEGs in the grafted and control samples during stress. The *x* axis shows logarithm fold change (Log2FC [Grafting/Control]) and *y* axis shows –Log10 (*p*-value). (**e**) Bar plot showing GO enrichment analysis (Panther, FDR < 0.05) for biological process in DEGs by the grafted samples. Red and blue color indicate higher and lower gene expression on significant DEGs (Adj. *p* < 0.05, Log2FC ≥ |1|). The *x* axis shows the –Log10 (*p*-value) categorized in each GO term (*y* axis).

**Figure 5 plants-11-01947-f005:**
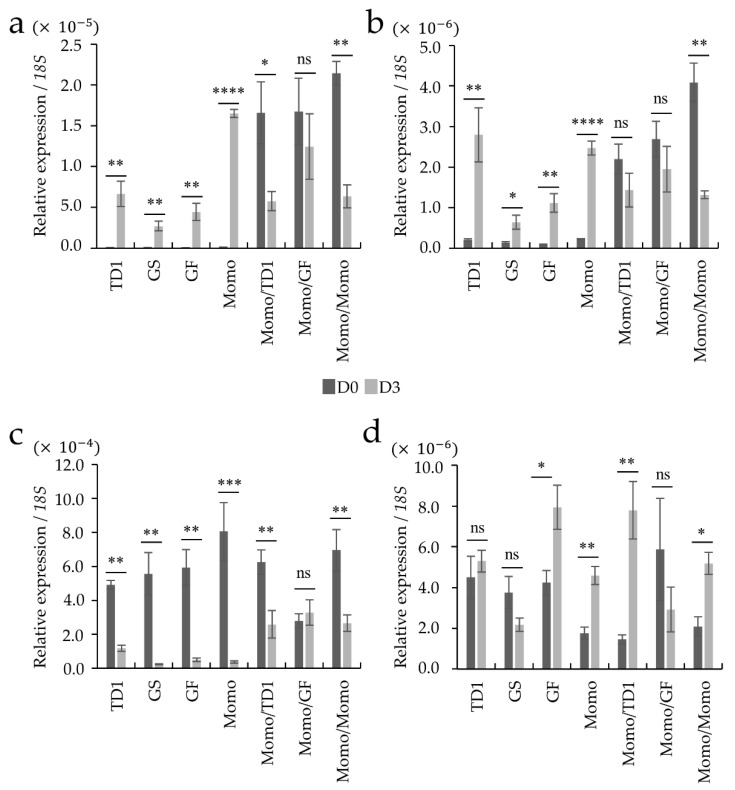
Changes in gene expression of 4 genes differentially expressed by grafting or drought stress assessed by quantitative real-time PCR. (**a**) Heat shock protein (*Solyc08g062450*); (**b**) Ascorbate peroxidase (*Solyc09g007270*); (**c**) Chlorophyll a–b binding protein (*Solyc08g067330*); and (**d**) 9-cis-epoxycaratenoid dioxygenase (*Solyc07g056570*). Gray bars represent before treatment (D0) and black bars represent 3 days drought stress treatment (D3). Asterisks denote significant differences according to t-test (Wilcoxon test) between D0 and D3 of each line, where * indicate *p* < 0.05, ** *p* < 0.01, *** *p* < 0.001, and **** *p* < 0.0001. No significance is denoted as *ns*. Error bars represent the standard error from 3 biological replicates with 3 technical replicates.

**Figure 6 plants-11-01947-f006:**
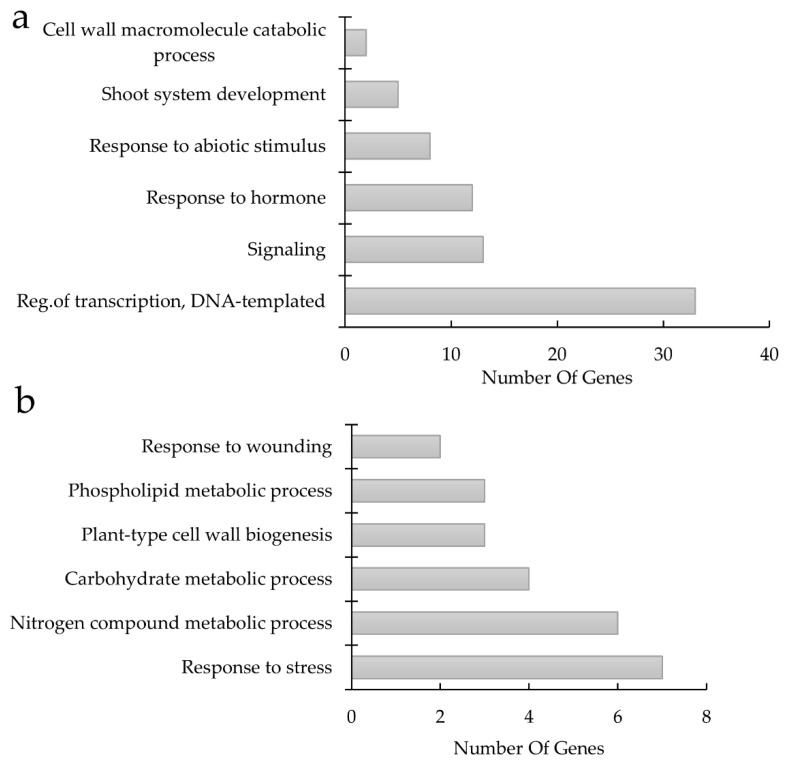
Compatible [Momotaro shoot (Momo)/TTM-079 (TD1) and Momo/Momo] and incompatible grafted lines [Momo/green force (GF)] show differentially expressed genes revealed by transcriptomic analysis. Gene ontology (GO) analysis classification on (**a**) reduced gene expression on DEGs by Momo/GF (fold change ≥ 2, Momo/TD1, Momo/Momo vs. Momo/GF) and (**b**) increased gene expression on DEGs by Momo/GF (fold change ≤ 0.5, Momo/TD1, Momo/Momo vs. Momo/GF). The *x* axis shows the number of genes of each GO cluster classification, and the *y* axis shows the GO ontology term.

**Table 1 plants-11-01947-t001:** List of hormone-related genes differentially expressed by grafting before drought stress (D0).

Gene ID	Annotation	Log2FC * (Grafted /Control)	Adj. *p*-Value
**Abscisic Acid-related genes**			
Solyc03g095780	Abscisic acid receptor PYL4-like	4.2	4.2 × 10^−3^
Solyc09g015380	Abscisic acid receptor PYL5	3.2	4.2 × 10^−3^
Solyc06g050500	Abscisic acid receptor PYL4-like	3.2	9.6 × 10^−3^
Solyc10g076410	Abscisic acid receptor PYL6	3.0	2.0 × 10^−2^
Solyc10g085310	Abscisic acid receptor PYL5	2.4	5.5 × 10^−3^
Solyc12g055990	Abscisic acid receptor PYL10	1.2	4.4 × 10^−3^
**Auxin-related genes**			
Solyc07g042490	Auxin-responsive SAUR protein	3.7	2.5 × 10^−3^
Solyc02g037550	Auxin efflux carrier	2.6	4.8 × 10^−3^
Solyc01g110580	Small auxin up-regulated RNA5	1.4	5.2 × 10^−3^
Solyc02g077560	Auxin response factor 3	1.3	4.7 × 10^−4^
Solyc01g110920	Small auxin up-regulated RNA26	1.3	2.9 × 10^−2^
Solyc09g007810	Auxin Response Factor 16A	1.0	2.0 × 10^−3^
Solyc05g046320	Small auxin up-regulated RNA55	−1.1	1.3 × 10^−2^
Solyc06g075150	Auxin Response Factor 10B	−1.5	4.3 × 10^−3^
Solyc01g099840	Dormancy/auxin associated protein	−1.6	1.6 × 10^−2^
Solyc02g077880	Auxin-repressed protein	−2.6	1.2 × 10^−3^
**Cytokinin-related genes**			
Solyc06g075090	Cytokinin riboside 5-monophosphate phosphoribohydrolase	1.7	3.7 × 10^−3^
Solyc04g080820	Cytokinin oxidase 4	1.4	4.6 × 10^−2^
Solyc10g082020	Cytokinin riboside 5-monophosphate phosphoribohydrolase	1.2	2.4 × 10^−2^
Solyc03g007460	*Solanum lycopersicum* Cytokinin Response Factor 4	−1.5	1.6 × 10^−2^
**Ethylene-related genes**			
Solyc04g011440	Ethylene-responsive heat shock protein cognate 70	5.4	8.6 × 10^−4^
Solyc09g059390	Ethylene-responsive transcription factor	4.9	1.8 × 10^−2^
Solyc12g056590	Ethylene Response Factor D.2	3.5	1.4 × 10^−2^
Solyc01g090300	Ethylene-responsive transcription factor 2	2.4	3.3 × 10^−2^
Solyc06g065820	Ethylene Response Factor H.1	2.0	4.5 × 10^−2^
Solyc01g106820	Ethylene-dependent gravitropism-deficient and yellow-green-like 3	1.9	9.6 × 10^−3^
Solyc08g066660	Ethylene-responsive transcription factor TINY	1.8	3.8 × 10^−2^
Solyc04g012050	Ethylene-responsive transcription factor	1.7	2.6 × 10^−2^
Solyc03g118190	Ethylene-responsive transcription factor	1.7	4.4 × 10^−2^
Solyc03g093610	Ethylene response factor A.2	1.6	3.0 × 10^−3^
Solyc01g006650	ETHYLENE INSENSITIVE 3-like 3 protein	1.5	1.0 × 10^−2^
Solyc04g078640	Ethylene-responsive transcription factor RAP2-1	−1.3	3.9 × 10^−2^
Solyc07g054220	Ethylene-responsive transcription factor	−1.5	2.4 × 10^−2^
Solyc06g019200	Ethylene-dependent gravitropism-deficient and yellow-green-like 2	−1.7	1.8 × 10^−2^
Solyc08g007230	Ethylene-responsive transcription factor 2	−1.8	4.9 × 10^−2^
Solyc02g088310	Ethylene-responsive transcription factor	−2.0	1.1 × 10^−2^
**Gibberellin-related genes**			
Solyc02g083880	Gibberellin-regulated protein 3	4.2	4.3 × 10^−4^
Solyc01g150176	Gibberellin-regulated protein	3.9	4.8 × 10^−4^
Solyc03g113910	Gibberellin-regulated protein 14	2.9	2.7 × 10^−3^
Solyc12g042500	Gibberellin-regulated family protein	1.8	3.3 × 10^−2^
Solyc09g074270	Gibberellin receptor	1.5	7.6 × 10^−3^
**Jasmonic acid-related genes**			
Solyc01g103595	Jasmonate ZIM domain protein l	2.0	4.2 × 10^−2^

* Logarithm fold change between grafted and control (Log_2_ FC > |1|, Adj. *p*-value < 0.05).

**Table 2 plants-11-01947-t002:** List of differentially expressed genes with potential significance to identify compatible grafts [Momotaro (Momo)/TTM-079 (TD1) and Momo/Momo] vs. incompatible [Momo/green force (GF)] lines.

Gene ID	Annotation	Log_2_FC * (Compatible vs. Incompatible)	
		Momo/TD1 vs. Momo/GF	Momo/Momo vs. Momo/GF
**GO:0006355 Regulation of transcription, DNA-templated**			
Solyc04g079930	Histone deacetylase complex subunit	1.0	1.3
Solyc07g008540	Zinc finger protein CONSTANS-LIKE 2	1.2	1.8
**GO:0009725 Response to hormone**			
Solyc05g047590	Pectinesterase	1.3	1.8
Solyc06g050500	Abscisic acid receptor PYL4-like	1.2	1.6
**GO:0009628 Response to abiotic stimulus**			
Solyc08g061130	Transcription factor HY5	1.1	1.1
Solyc09g065660	Heat shock transcription factor	1.6	3.6
**GO:0006950 Response to stress -> GO:0042594 Response to starvation**			
Solyc02g088240	Phosphate transporter PHO1-like protein 5	−2.6	−2.9
Solyc08g068240	Phosphate transporter PHO1-like protein	−1.2	−1.7
**GO:0005975 Carbohydrate metabolic process -> GO:0009832 Plant-type cell wall biogenesis**			
Solyc01g103860	COBRA-like protein	−1.2	−1.0
Solyc03g114900	COBRA-like protein	−2.1	−3.0
**GO:0006950 Response to stress -> GO:0009611 Response to wounding**			
Solyc08g036660	Jasmonate ZIM domain protein f	−4.7	−3.6
Solyc08g036640	Jasmonate ZIM domain protein f	−3.1	−2.4

* Logarithm fold change between compatible and incompatible (Log_2_FC > |1| RPKM).

## Data Availability

The data of this study are available from the authors upon reasonable request. All global gene expression data are deposited in the BioProject database as the accession number of PRJDB13627.

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
