# Peer review of "Comparative Transcriptome Analysis of Grafted Tomato with Drought Tolerance"

_plants, 2022, doi:10.3390/plants11151947_

Round 1

Reviewer 1 Report

The authors present a description of the effects of grafting on the drought stress and tried to identify key genes involved in this process. For the most part, the work is well done, and very interesting. However, the work is fundamentally descriptive, still have to improve. First, it is really intriguing that both of Mono and TD1 did not show strong resistance to drought but the grafting line Mono/TD1 displayed more resistance to drought, please describe why in the results or discussion part? Second, the author analyzed the transcriptome and identified some gene very possibly involved in this phenotype, but the author needs to verify the RNA-seq data before concluding. Overall the manuscripts is very interesting and helpful in the future for agriculture

Reviewer 2 Report

This manuscript is a genetic analysis of the drought tolerance characteristics of tomato grafting.

The content of the introduction and discussion should be improved a lot.

Also, describe your materials and methods in more detail so that other researchers can reproduce them.

For more details, please refer to the attached file and modify it.

Round 2

Reviewer 1 Report

The revised manuscripts looks great and I do not question for this version and happy to see it publication.

Reviewer 2 Report

I judged that the authors have sufficiently revised the contents of the manuscript.

However, it would be better to put an underscore at the end of the form of table 1 like table 2.